# Survey of awareness of radiation disasters among firefighters in a Japanese prefecture without nuclear power plants

Koji Yamada[1,2,3], Izumi Yamaguchi[1,3], Hideko Urata[1,4], Naomi Hayashida[1,3]*

1 Division of Disaster and Radiation Medical Sciences, Nagasaki University Graduate School of Biomedical Sciences, Nagasaki, Japan, 2 Nagasaki City Fire Department, Nagasaki, Japan, 3 Division of Strategic Collaborative Research, Atomic Bomb Disease Institute, Nagasaki University, Nagasaki, Japan, 4 Department of Molecular Medicine, Atomic Bomb Disease Institute, Nagasaki University, Nagasaki, Japan

* naomin@nagasaki-u.ac.jp

**Data Availability Statement:** Data cannot be shared publicly because of inclusion of individual information. Data are available from the Ethics Committee (contact via Nagasaki University) for

## Abstract

Japanese firefighting organisations are essentially run as village, town, or city units. The Great Hanshin Earthquake of 1995 led to the establishment of emergency support teams to ensure rapid action in response to disasters beyond the capacities of local fire departments. The 2011 Great East Japan Earthquake involved both a tsunami and a radiation disaster caused by a nuclear reactor meltdown, underscoring the need for responses in complex disasters. This study aimed to assess Nagasaki Prefecture firefighters' preparedness for, awareness of, and anxiety regarding radiation disaster response with the aim of elucidating the factors affecting individuals' decisions to accept or reject assignment to a radiation disaster response team. A questionnaire survey was carried out with 1,122 firefighters in three firefighting departments in Nagasaki Prefecture, which does not have nuclear power plants. In total, 920 questionnaires were returned, and the 784 that were valid were analysed. Among the participants, 39% replied that they would have no difficulty accepting assignment to a radiation disaster response team; most of them were under 30 years old and unmarried. This group also included significantly higher percentages of participants who were confident about radiation disaster response or, if anxious, believed things would turn out fine, as well as those who replied that they were able to use the necessary equipment. Furthermore, this group had significantly higher percentages of participants who replied that they would definitely participate in seminars and those who replied that their level of preparedness for radiation disasters was sufficient. The willingness to be assigned to a radiation disaster response team was linked to confidence about radiation disaster response and about handling materials and/or equipment. Therefore, it is considered that measures to increase firefighters' confidence regarding response to radiation disasters will be linked to more proactive measures if and when such disasters occur.

researchers who meet the criteria for access to confidential data. Requests for the data set require permission from the Ethics Committee at Nagasaki University Graduate School of Biomedical Sciences. The contact information for the ethics committee is as follows: Tel.: 095–819–7198; Website: http://www.mdp.nagasaki-u.ac.jp/research/support_rinri.html.

**Funding:** The authors received no specific funding for this work.

**Competing interests:** The authors have declared that no competing interests exist.

## Introduction

In Japan, the rationale for firefighter activities is stipulated in Clause 1 of the Fire Organization Law, as follows [1]:

Firefighting involves the use of facilities and personnel in order to protect the lives, physical health, and property of citizens against loss due to fires; prevent disasters due to fires, floods, earthquakes, and so on; reduce injuries and other harm caused by such disasters; and carry out appropriate activities for the transport of persons injured by disasters, and so on.

On the basis of this law, firefighters are expected to respond promptly to the various types of disasters that occur in Japan.

In Japan, earthquakes, typhoons, and cloudbursts are an annual occurrence, and in recent years, disasters have become especially complex and diverse. For example, disasters in the recent past include the avalanche in Nasu, Tochigi Prefecture, on 27 March 2017, in which eight people died and 40 injured, and the volcanic eruption of Mt. Ontake on 27 September 2014, in which 58 people died, 69 were injured, and five remain missing. There have also been disasters in built-up places, such as the warehouse fire in Miyoshi, Saitama Prefecture, on 16 February 2017, which took firefighters 13 days to extinguish. Then, there was the West Japan Railway Co.'s Fukuchiyama Line derailment, which happened in an urban area on 25 April 2005 and resulted in as many as 107 deaths and 549 injuries. Finally, a fire at Bridgestone's Tochigi Plant, in Kuroiso, Tochigi Prefecture, on 26 September 2003, resulted in 5,032 people from 1,708 households in seven wards being evacuated for two days. Thus, disasters that are large-scale in terms of severity and the number of people affected occur quite frequently.

Climate change is leading to more severe natural disasters, and urban development is leading to increasingly complex and diverse disasters that necessitate a specialised approach to management. Firefighters are expected to respond to these promptly and therefore prepare for them on a daily basis.

In accordance with the relevant laws, firefighting organisations in Japan operate as city, town, and village units. In one of these laws, the role of municipalities is stated as follows:

To ensure such continual and steady improvement and strengthening of the fire service strength, municipalities must perform certain tasks while implementing the strict safety management of fire personnel. These tasks include (i) improvement and strengthening of the fire suppression system through the advancement of fire defence tactics and fire equipment so that various types of disasters can be dealt with in a precise manner, (ii) improvement and strengthening of the fire prevention system in response to the higher demand for and specialisation of fire prevention work which have resulting from the growing size and complexity of buildings, (iii) improvement and strengthening of the ambulance service system in response to the growing number of ambulance runs and demand for a higher quality ambulance service which have resulted from the progress of the ageing society, (iv) improvement and strengthening of the rescue system to conduct rescue operations in a precise manner at the time of disasters of which the complexity and diversity have been growing and (v) improvement and strengthening of the system to implement measures designed to protect the public in an armed attack and other situations. Moreover, consolidation of a wide area fire service system, including emergency fire response teams, is required to strengthen the preparedness against such large-scale natural disasters as earthquakes, violent storms and floods. [2]

Firefighters work at all fire departments run by city, town, and village governments. After learning firefighting fundamentals (e.g., fire suppression, first aid, rescue, and prevention) at fire and disaster management colleges, which are mostly run by prefectural governments, firefighters are employed within organisations defined by each fire department in accordance with the relevant city, town, or village's population, area, types of buildings, and the like. The fire department's main categories of tasks are fire suppression, first aid, rescue, communication, and prevention. Concurrent performance of duties is permitted, depending on each department's size and circumstances.

Because of the Great Hanshin Earthquake on 17 January 1995, emergency firefighting support teams, along with an organisation for mutual support between firefighting bodies throughout Japan, were established in June 1995. These teams were legally established in April 2004 by an amendment to the Fire Organization Law and have since been able to respond to a wide variety of disasters [3]. When a fire disaster occurs, the governor of the prefecture where the disaster is located submits a mobilisation request to the commissioner of the Fire and Disaster Management Agency, who then submits dispatch requests to the governors of prefectures and/or mayors of cities, towns, and villages other than where the disaster has occurred. The fire departments that receive these requests then marshal and mobilise emergency firefighting support teams, as prefecture-based units, and thus actively respond to the disaster at the site where it has occurred. Therefore, each fire department, based on instructions from the relevant prefectural government, records the teams and personnel that can be dispatched in the event of a request and prepares for such a request. In terms of selection of teams to dispatch, individuals and teams are designated in accordance with conditions in each fire department such as the department size.

Among the disasters to which the emergency firefighting support teams have responded, the Great East Japan Earthquake on 11 March 2011, was one such as had not previously occurred in Japan. In addition to the earthquake and tsunami, there was a radiation disaster caused by an incident at Tokyo Electric Power Co.'s Fukushima Daiichi Nuclear Power Plant (hereafter 'Fukushima Daiichi Disaster'). Firefighting organisations throughout Japan mobilised as many as 31,166 emergency support teams consisting of 109,919 personnel, and activities at the disaster site continued for 88 days [4]. In addition, with respect to the Fukushima Daiichi Disaster, as well as the firefighters at the fire department responsible for that location, nine other fire departments dispatched a total of 136 emergency support teams, for a total of 664 personnel. Tokyo Electric Power Co. also hired 12 firefighting pump engines from eight fire departments [5].

After the establishment of the emergency firefighting support teams, from June 1995 to September 2017, they went into action 34 times, and there have been multiple mobilisations a year since 2014. It is expected that situations in which firefighters are active at disaster sites performing duties outside their usual ones, as in the case of the Great East Japan Earthquake, will continue to increase in frequency. The Fukushima Daiichi Disaster was the first large-scale radiation disaster in Japan. There had been no previous incidents that involved a risk of major release of radioactive substances, and it constituted a major turning point for firefighters in terms of the appropriate activities in the event of a radiation disaster. The 2018 White Paper on Fire Service [6], released after the Fukushima Daiichi Disaster, mentions future measures to be taken based on the disaster, referencing how certain health interventions, for example tests using whole-body counters, were given to firefighters on the emergency firefighting support teams involved in responding to the disaster to help reassure them. In addition, health management support was provided, including ensuring opportunities for periodic supplementary tests and long-term monitoring of health conditions among firefighters. Thus, the importance of providing continuing support was recognised. In addition, the paper stated that, in

the event of accidents involving release of radiation or radioactive materials that occur either (i) at nuclear energy facilities or facilities where radioisotopes and similar materials are handled or (ii) when transporting radioactive materials, it is essential for the relevant firefighting organisations to respond promptly and appropriately. Thus, it is also essential to continue to improve the firefighting skills of these organisations in relation to accidents involving the release of radioactive materials [6].

From the point of view of firefighters, who must confront various types of disasters, elucidation of the current situation regarding firefighting related to radiation emergencies and the factors affecting timely responses to radiation disasters could inform how to improve preparation for radiation disasters. Previous studies reported the effectiveness of education and training for disaster preparedness or planning [7–10]. Furthermore, Ben Natan et al. examined variables affecting nurse willingness to work at an earthquake site and reported that 'perceived self-efficacy, level of knowledge, experience and the support of a multidisciplinary staff affect nurse willingness to report to work in the event of an earthquake' [11]. Radiation disasters are very rare but require specialization. A report about the special disaster, that is, the chemical, biological, radiological and nuclear (CBRN) events reported that "CBRN events are probably more of a means of disorganization and major terror than of mass destruction. During crisis situations, decision-making is most effective when the simplest and most effective ideas are used" [12]. However, there has been no research on disaster preparedness or willingness to work in disasters among firefighters. By eliminating the anxiety of firefighters, cooperation with related parties will be smooth, and firefighters can engage efficiently with peace of mind while ensuring safety. In this context, a survey was carried out with firefighters in Nagasaki Prefecture, which does not have nuclear power plants. The aim was to assess the preparation for, awareness of, and anxiety regarding responses to radiation disasters, with the aim of determining the factors influencing individuals' decisions to accept or refuse assignment to a radiation disaster response team when this possibility is proposed.

## Materials and methods

This survey was carried out from November to December 2016 with 1,122 firefighters at three fire departments in Nagasaki Prefecture and covered all personnel in each fire department. These personnel engage in five tasks: fire suppression; first aid; rescue; communication; and prevention, and sometimes they have overlapping duties. These departments included full-time and specialist rescue teams, as well as full-time staff in command roles, so that few personnel had overlapping duties. Although the three fire departments do not have a specialised section for unique disasters such as those involving nuclear materials, chemicals, or biohazards, they do have personnel engaged in communication and prevention who may be called on to work in these kinds of disasters.

We conducted a questionnaire survey with all participants. We delivered the explanatory documents about the survey and the answer sheets to the fire departments in the same envelopes, and then distributed them to all respondents. Respondents placed the completed answer sheets in the same envelopes used for distribution. The fire departments collected these, which were received by the researchers. We understood returning a completed questionnaire as providing consent to participate in the study. Participants were allowed to leave questions blank if they did not wish to provide an answer; completely blank questionnaires were discarded. Partially completed questionnaires were also accepted and included in this study.

The principal contents of the questionnaire were respondents' attributes, awareness about the effects of nuclear incidents, and awareness about radiation disasters. To explain the above in more detail, on the basis of their attributes, participants were categorised by age, marital

status, duration of service, details of duties, experience, and number of mobilisations to large-scale and radiation disasters.

The items relating to awareness about the effects of nuclear incidents were as follows: interest in events at the time of nuclear incidents, awareness of the effects of nuclear incidents on participants' own health, awareness of the effects of nuclear incidents on the health of adults and children, and awareness of genetic effects.

The items relating to awareness about radiation disasters were as follows: potential for a nuclear disaster in the area where one is stationed, level of confidence in responding to radiation disaster, understanding of the details of firefighting teams' activities at the time of the Fukushima Daiichi Disaster, experience of attending lectures about radiation disasters and/or emergency exposure situations, participants' own competence at using the materials and/or equipment for radiation disasters, understanding of the manual, wish to participate in future seminars about radiation disasters, participants' response if voluntary transfer to a team that is mobilised preferentially for radiation disasters is proposed, views about voluntarily working in an area where the air radiation dose rate has been measured to be 100 mSv/year, if this is proposed, and participants' own preparedness for future radiation disasters (see Supplementary Materials of S1 Data for specific details of questions and responses).

Of the 1,122 firefighters who responded, it was confirmed from their names that 14 were women. However, as this number was small and there were concerns about identifying individuals from the contents of the replies, a question concerning gender was not included.

In relation to the replies to the questions about attributes, awareness about the effects of nuclear incidents, and awareness about radiation disasters, the factors affecting the willingness to be assigned to a radiation disaster response team were analysed using the $\chi^2$ test and ordinal logistic regression analysis. The software used for statistical analysis was SPSS (version 24; IBM). The significance level was set at $p<0.05$.

In accordance with the *Ethical Guidelines for Medical and Health Research Involving Human Subjects* published jointly by the Japanese Ministry of Education, Culture, Sports, Science, and Technology and the Ministry of Health, Labour, and Welfare, this study was approved by the Ethics Committee at Nagasaki University Graduate School of Biomedical Sciences (approval no.: 16092399).

## Results

Of the 1,122 questionnaires distributed, 920 were returned (response rate: 81.9%). The number of blank and incomplete questionnaires was 15 (1%) and 121 (10%), respectively. After excluding these, 784 valid questionnaires (valid response rate: 70%) were analysed.

The distribution of participants by age group was as follows: under 30: 31%; 30–39: 21%; 40–49: 19%; 50–59: 21%; and 60 and over: 8%. Married and unmarried participants corresponded to 76% and 24%, respectively. The distribution of different durations of service was as follows: less than six years: 21%; 6–10 years: 19%; 11–20 years: 16%; 21–30 years: 16%; 31–40 years: 21%; and over 40 years: 7%. The percentages of different duties were as follows: mainly fire suppression duties: 50%; first-aid duties: 21%; routine administrative duties: 13%; rescue duties: 12%; and command duties: 4% (Fig 1). Participants with and without experience of mobilisation to large-scale disasters corresponded to 27% and 73%, respectively. Of those with experience, the percentages of mobilisations were as follows: once: 77%; twice: 10%; thrice: 2%; and four times or more: 11%. Only one participant reported having been mobilised for a nuclear incident or radiation disaster.

The percentage of participants who replied that they would accept assignment to a team that would be mobilised preferentially in the event of a radiation disaster without hesitation

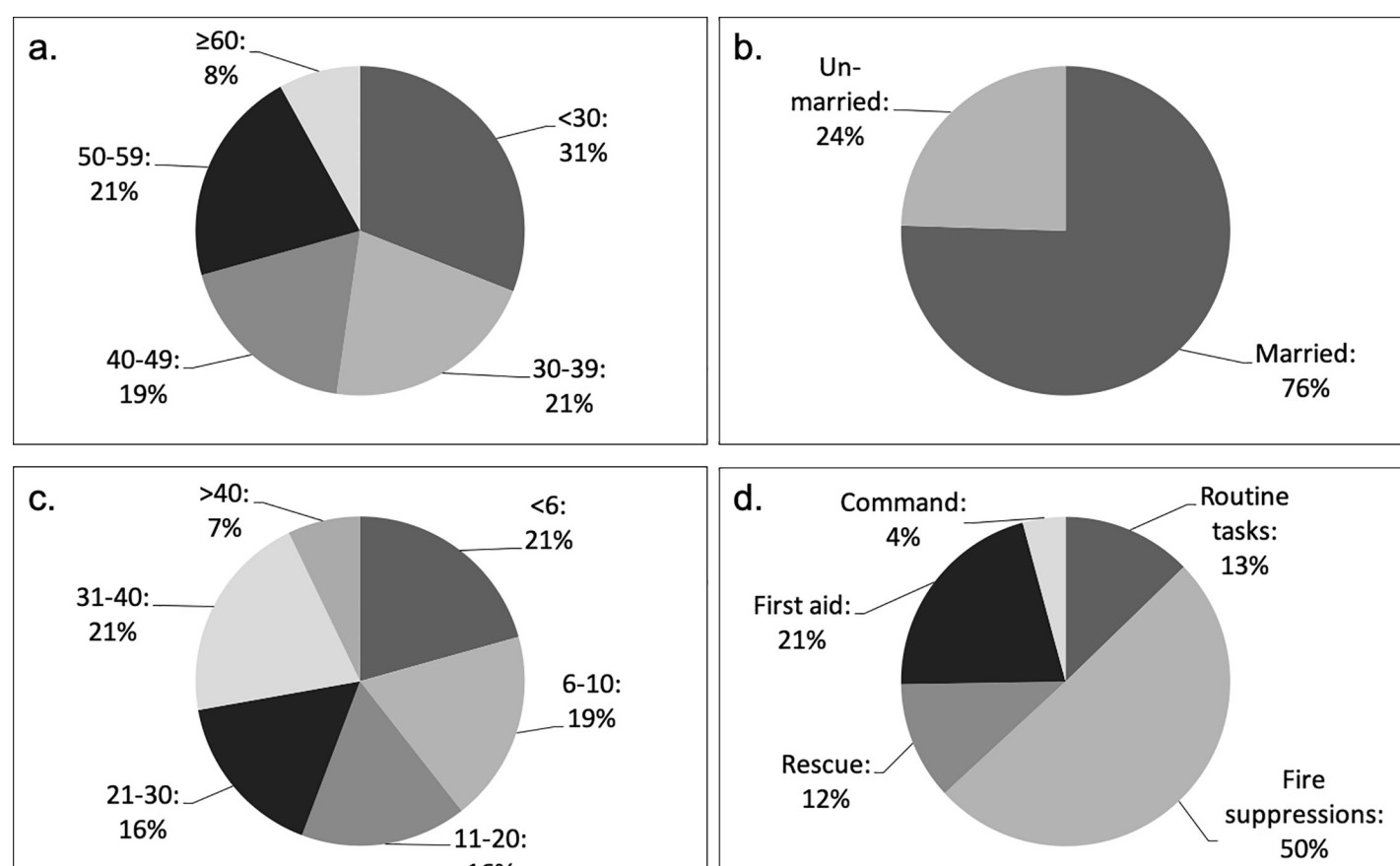

**Fig 1. Attributes of participants.** (a) Age (years), (b) Marital status, (c) Duration of service (years), (d) Details of activities.

was 39% (Fig 2A). We analysed factors associated with willingness to accept assignment without hesitation using the $\chi^2$ test. Significant correlations were found with age, marital status, awareness of the health and/or genetic effects of nuclear incidents, confidence about response to radiation disasters, competence in using materials and/or equipment for radiation disasters, future willingness to participate in seminars, and whether the participant's preparedness for a radiation disaster was sufficient (Tables 1 and 2). More participants aged under 30/in their 30s (p<0.001) or unmarried (p = 0.004) replied that they would accept assignment to a team that would be mobilised preferentially in the event of a radiation disaster without hesitation than other participants, and this result was significant. Regarding awareness about the effects of nuclear incidents, more participants who considered that nuclear incidents do not affect health (p<0.001) or genetics (p<0.001) replied that they would accept assignment without hesitation, which was also significant. In the analysis of awareness of radiation disasters, a significantly high percentage of participants who would unhesitatingly accept assignment answered that they thought the potential for a nuclear disaster in the area where one is stationed was possible (p = 0.011), and they had confidence, even if they had anxiety, in responding to radiation disasters (p<0.001). Furthermore, regarding participants' own competence at using materials and/or equipment for radiation disasters, there was a significant relationship between participants who answered 'I can use them' and those who were willing to accept assignment without hesitation (p = 0.016). For education and preparation, there was a significant relationship

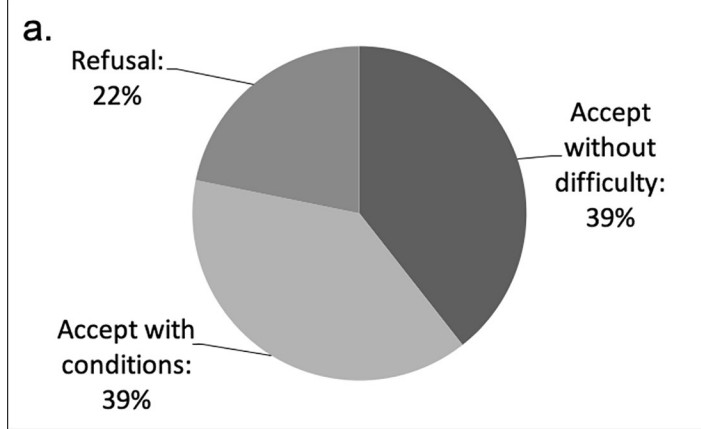

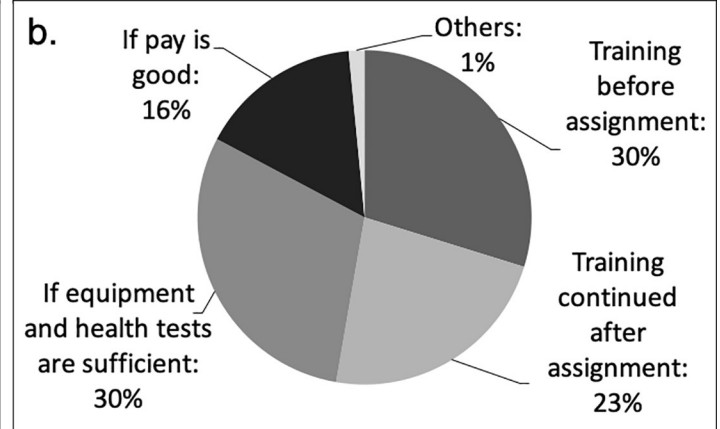

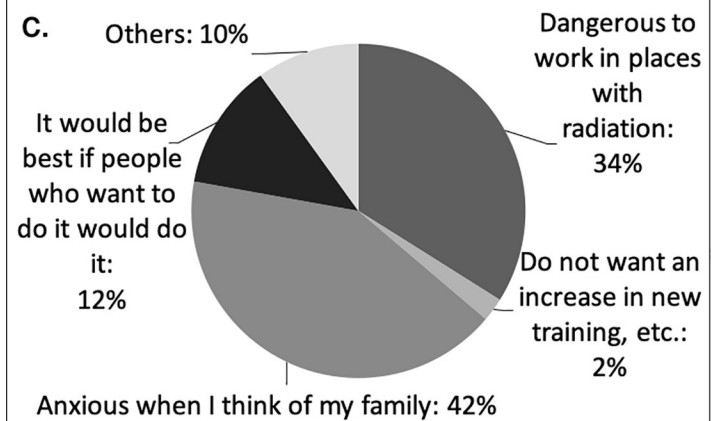

**Fig 2. Opinion about radiation disasters.** (a) Whether I would accept an assignment with a radiation disaster response team. (b) Conditions under which I would accept assignment. (c) Reasons for refusing assignment.

between participants who would definitely participate in future seminars about radiation disasters (p<0.001) and those who were sufficiently prepared for future radiation disasters (p<0.001) and accepting assignment without hesitation.

Ordinal logistic regression analysis of factors with independent relationships with acceptance of assignment to radiation disaster response teams revealed that participants who answered as follows were significantly more likely than those who gave other answers to accept

**Table 1. Factors correlated with opinion about assignment to radiation disaster response team: Attributes and awareness about the effects of nuclear incidents.**

| Variable | Accept without hesitation N = 309 (%) | Accept with conditions N = 304 (%) | Refuse N = 171 (%) | P-value |
|---|---|---|---|---|
| Age (<30/30s/40s/50s/ ≥60) | 115/77/50/60/7 (37/25/16/19/2) | 88/64/69/66/17 (29/21/23/22/17) | 40/26/25/41/39 (23/15/15/24/23) | <0.001 |
| Married/unmarried | 215/94 (70/30) | 236/68 (78/22) | 141/30 (82/18) | 0.004 |
| Nuclear incidents have health effects* (yes/no) | 264/45 (85/15) | 293/11 (96/4) | 162/9 (95/5) | <0.001 |
| Nuclear incidents have genetic effects* (yes/no) | 241/68 (78/22) | 273/31 (90/10) | 150/21 (88/12) | <0.001 |

*Yes: 'I think there are effects' or 'If I have to choose, I think there are effects'; No: 'If I have to choose, I think there are no effects' or 'I think there are no effects'.

**Table 2. Factors correlated with opinion about assignment to radiation disaster response team: Awareness about radiation disasters.**

| Variable | Accept without difficulty N = 309 (%) | Accept with conditions N = 304 (%) | Refuse N = 171 (%) | P-value |
|---|---|---|---|---|
| Possibility of radiation disaster at station (yes/no) | 201/108 (65/35) | 214/90 (70/30) | 97/74 (57/43) | 0.011 |
| Confidence about response to radiation disaster (confident/anxious/not confident) | 133/176 (43/57) | 79/225 (26/74) | 26/145 (15/85) | <0.001 |
| Activities of firefighting team at time of nuclear incident (aware/unaware) | 243/66 (79/21) | 244/60 (80/20) | 124/47 (72/28) | 0.138 |
| Attend lectures about radiation disasters, etc. (yes/no) | 151/158 (49/51) | 157/147 (52/48) | 69/102 (40/60) | 0.058 |
| Using materials and/or equipment for radiation disasters* (can use/not confident/cannot use) | 78/161/70 (25/52/23) | 55/179/70 (18/59/23) | 22/77/72 (13/45/42) | <0.001 |
| *Manual for Firefighting Activities at Nuclear Power Facilities*** (familiar or comprehensible/incomprehensible) | 98/211 (32/68) | 98/206 (32/68) | 39/132 (23/77) | 0.068 |
| Participation in seminars about radiation disasters*** (will participate/will consider/will not participate) | 73/202/34 (24/65/11) | 65/201/38 (21/66/13) | 13/91/67 (7/53/40) | <0.001 |
| Preparedness for radiation disasters (fully prepared/partly prepared/unprepared) | 9/183/117 (3/59/38) | 1/148/155 (0/49/51) | 2/59/110 (1/35/64) | <0.001 |

*Can use: able to use the materials and/or equipment; not confident: understand their details and location; cannot use: not able to use them.

**Familiar: fully comprehensible; understood: largely comprehensible.

***Will participate: will definitely participate; will consider: participation would be acceptable; will not participate: will probably not participate, or will not participate.

assignment without hesitation: (i) 'I am confident about dealing with radiation disasters' (p<0.001), (ii) 'I can use the relevant materials and/or equipment' (p = 0.016), and (iii) 'I will definitely participate in future seminars' (p = 0.001) (Table 3).

Regarding participants who replied that they would accept assignment to a radiation disaster response team under certain conditions, the results were as follows: sufficient education and training before assignment: 30%; satisfactory personal equipment and health checks: 30%; continued education and training: 23%; and good pay: 16%. Reasons for refusal of assignment to a radiation disaster response team were the following: anxiety owing to family: 42%; danger associated with radiation disasters: 34%; the belief that people must willingly work under such circumstances: 12%; and aversion to being subjected to more education and training: 2% (Fig 2B and 2C).

With respect to the willingness to work where the radiation dose rate in the air has been measured to be 100 mSv/year, 54% of participants replied that they would agree if proposed. The different reasons for this willingness were as follows: being a firefighter: 82%; that 100 mSv/year has no effect on health: 7%; and inability to refuse: 8%. The different reasons for refusing to work in such an area were as follows: 100 mSv/year would harm health: 16%; desire to avoid radiation exposure out of concern for the family: 35%; and anxiety owing to lack of experience with radiation disasters: 38% (Fig 3).

**Table 3. Independent factors correlated with opinion about assignment to radiation disaster response teams.**

| | B | 95% confidence interval | | |
| | | Lower limit | Upper limit | Significance level |
|---|---|---|---|---|
| Nuclear incidents have health effects (yes/no) | -0.902 | -1.942 | 0.139 | 0.089 |
| Confidence about response to radiation disasters (confident/anxious/not confident) | 0.829 | 0.477 | 1.180 | <0.001 |
| Using materials and/or equipment for radiation disasters (can use/not confident/cannot use) | 0.291 | 0.055 | 0.528 | 0.016 |
| Participation in seminars about radiation disasters (will participate/will consider/will not participate) | 0.688 | 0.285 | 1.091 | 0.001 |
| Preparedness for radiation disasters (fully prepared/partly prepared/unprepared) | 0.255 | -0.042 | 0.553 | 0.092 |

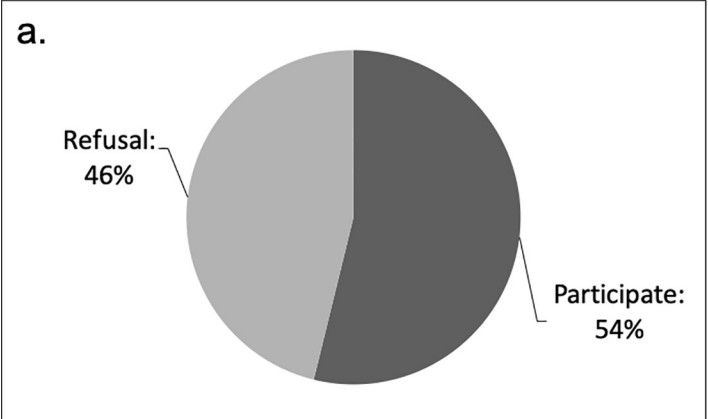

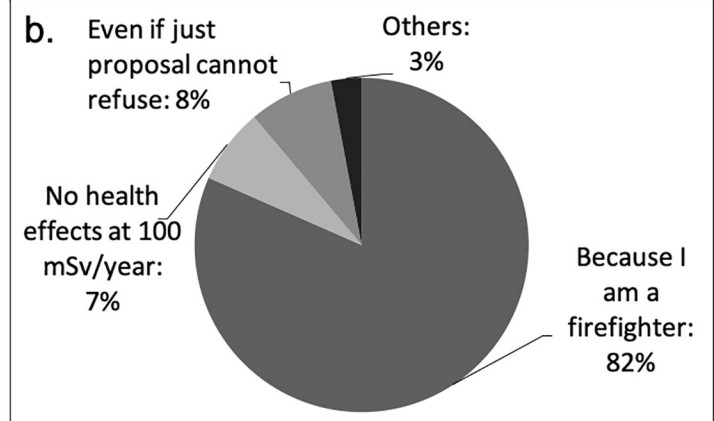

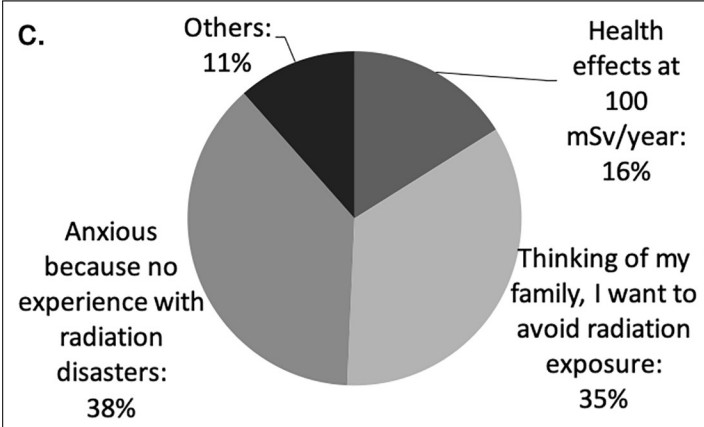

**Fig 3. Opinion about radiation disasters.** (a) Voluntarily engaging in activities in areas with radiation dose of 100 mSv/year, (b) Reasons for engaging in activities in areas with radiation dose of 100 mSv/year, (c) Reasons for refusing to engaging in activities in areas with radiation dose of 100 mSv/year.

With respect to the wish to participate in future seminars about radiation disasters, 80% of participants replied that they would definitely participate or be willing to participate, which was much greater than the 17% who replied that they would probably participate and the 3% who replied that they would not participate. The different reasons for not wanting to participate in seminars were as follows: infrequency of radiation disasters: 31%; being busy with other things: 29%; scepticism about the benefits of participation: 27%; and already having undergone education and training: 4% (Fig 4).

## Discussion

In this study, we found that 39% of firefighters in Nagasaki Prefecture would have no problem being assigned to a team that would be preferentially mobilised to respond to radiation disasters. Study participants who were confident about dealing with radiation disasters, able to use the relevant materials and/or equipment, and would participate in relevant seminars were significantly more likely than others to report a lack of hesitation regarding being assigned to such a team.

The Ministry of Internal Affairs and Communications has released several documents providing guidelines for firefighters responding to radiation disasters in Japan, including the *Manual for Firefighting Activities at Nuclear Power Facilities*, *Etc.* [13] and *Start*! *RI119*: *Basic*

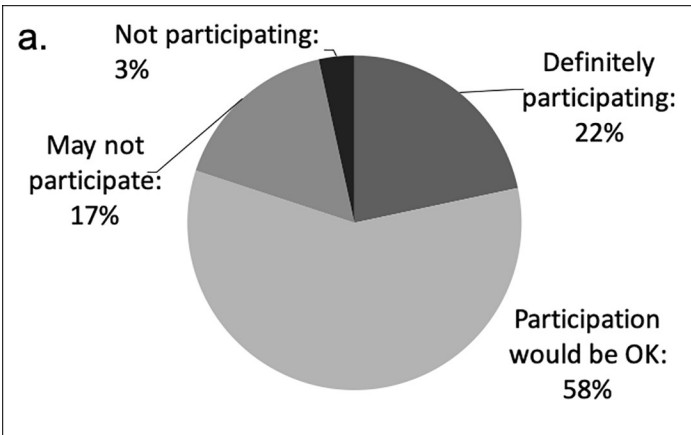
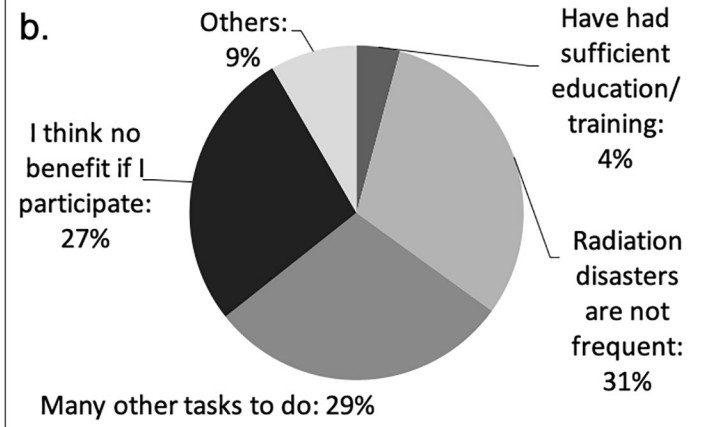

**Fig 4. Participation in seminars about radiation disasters.** (a) Wish to participate in seminars, (b) Reasons for not participating in seminars.

*Knowledge for Responses to Radiation Disasters by Firefighters* [14], and various fire departments have prepared their own manuals for responding to radiation disasters based on these. Fire academies in each prefecture have also planned educational programmes relating to radiation disasters, depending on the situation in each area, and currently invite specialists to hold lectures and provide practical training. In addition, in recent years, numerous books and other documents have been published about responding to various types of radiation disasters, and the opportunities for firefighters to become acquainted with these have increased. Furthermore, so that the materials and/or equipment they have can be used effectively, training based on the manuals is provided routinely, and the materials and/or equipment are upgraded as necessary. In our study, 42.8% of participants were familiar with or said that the *Manual for Firefighting Activities at Nuclear Power Facilities* was comprehensible. Ahayalimudin N et al. [15] reported almost all (93%) emergency medical workers agreed on the importance of reading and understanding their institution's disaster management plan, and the majority were willing to be involved in disaster management, but half felt they did not have the responsibility to assist disaster victims with their basic needs. In our data, there was no association between the frequency of participants who were familiar with the manual and their opinion about assignment to a radiation disaster response team, meaning that it is important to not only prepare and maintain a manual but also to conduct useful training with the manual. Globally, although there are differences in systems between countries in terms of governmental jurisdiction, actual tasks, and so on, firefighters respond to similar types of disasters, including radiation disasters, in each country and region [16–21]. For example, in the case of Norway, in addition to the existence of an emergency preparation protocol, training within the organisation is particularly important [22].

The results of the present study show that firefighters' activities when radiation disasters occur are affected by their level of confidence in their capacity to respond appropriately to the disaster. This confidence involves their pre-disaster experience as well as their skill at disaster response and handling the relevant materials and/or equipment. Regarding response to radiation disasters, participants who were confident or anxious but believed that things would turn out fine were significantly more likely to have no difficulty with assignment to a radiation disaster response team than others. However, 70% of participants replied that they were not confident; therefore, although firefighters who have to confront radiation disasters are provided with books and educational programmes, and training is carried out when materials

and/or equipment are upgraded, it is possible that these measures are not put into practice and do not lead to increased confidence.

Of all participants, 78% replied that they would either accept without hesitation or accept under certain conditions assignment to a radiation disaster response team if proposed. In addition, 54% of participants replied that they would accept if the possibility of voluntarily engaging in activities in areas with air radiation dose rates of 100 mSv/year were proposed.

A previous study [23] about the willingness of civil defence and firefighting programme students to work in disasters after graduation reported only 42% of students were willing to work in nuclear accidents. The study also reported 46.3% of the students 'wanted to work in disasters because they wanted to help people', and only 22.4% thought that difficult and uncomfortable conditions would reduce their willingness to work [23]. Therefore, even student firefighters have a sense of mission to save lives or help people. Considering that 82% of participants in our study gave 'being a firefighter' as the reason why they would accept assignment to a disaster response team, this willingness can be attributed to the fact that firefighters are habituated to confronting disasters and exemplifies the strength of their perception of their mission as requiring them to respond even to radiation disasters.

According to a radiation risk awareness survey [24] performed with firefighters in Aomori Prefecture, which does have nuclear energy facilities, the firefighters had few uncertainties or anxieties about radiation. However, it should be noted that personnel at fire departments in prefectures with nuclear energy facilities have raised concerns about the safety of disaster responses and facilities as well as insufficient knowledge and understanding of radiation by personnel at fire departments in prefectures without nuclear energy facilities. It is thus necessary to prepare a curriculum for specialists in radiation disasters, to improve the skills of individual personnel by means of lectures and seminars, and to increase knowledge and understanding so as to enable team-based disaster responses [24]. Another study [25] on factors affecting emergency medical workers' disaster response competency reported participation in disaster education or training two or more times annually was significantly associated with competency. It also reported disaster risk perception, self-efficacy for disaster, and personal disaster preparedness predicted disaster response competency. A systematic review of disaster preparedness among nurses found that previous disaster response experience and disaster-related training increase disaster response preparedness [26]. In fact, provision of high quality and carefully structured education and training courses and proactive participation in the seminars may be the most important factor for improving individual attitudes. In the present study, continuing education and training was put forward as a condition for accepting assignment to a radiation disaster response team, but some participants replied that they had doubts about the value of participating in seminars. However, irrespective of the willingness to accept a nuclear disaster assignment, the level of interest in participating in seminars about radiation disasters was high, and the importance of training for radiation disasters was recognised. This suggests that when seminars and continued training are provided, measures and approaches that enable individuals to raise their confidence about the appropriate responses and handling the relevant materials and/or equipment before a radiation disaster has occurred should be included.

Measures such as the above have already been attempted by Kobe Municipal Fire Dept. In October 2007, the agreement "Memorandum on Cooperation in Times of Specialized Disaster" [27] was signed by the Kobe Municipal Fire Dept. and Kobe Gakuin University; later, other parties also signed the memorandum, these being Kobe Pharmaceutical University in August 2008 [28] and Kobe University in March 2012 [29]. The memorandum's contents include (i) support and cooperation in relation to radiation and other specialised disasters; (ii) provision of education and training courses to firefighters, collaborative research, holding

public seminars, and so on, from before such disasters occur; and (iii) education of students about preventing fires and other disasters, with firefighters as lecturers [29–31]. Based on this memorandum, the education, training, and so on necessitated by either party's circumstances are to be provided. However, considerable work remains to be done to verify its effectiveness.

This study involved a questionnaire survey with firefighters in Nagasaki Prefecture, which does not have nuclear power plants. However, Nagasaki suffered from fallout from the atomic bomb in 1945. In 1943, during World War II, the national government established a policy of expanding the government-run firefighting organisations in all important Japanese cities. As a result, one fire station was opened in Nagasaki and one nearby in Sasebo; established within the national police organisation, these two fire stations were the first specialised, government-run firefighting organisations in Japan. However, as that was during the war, unlike the current situation, the firefighting activities focused on fires due to aerial bombardment. It was during that time that the atomic bomb was dropped on Nagasaki, at 11:02 on 9 August 1945, causing 73,884 deaths and 74,909 serious injuries, with 12 deaths and 26 injuries among firefighters. Of the 145 firefighters in Nagasaki at that time, 98 were left after deaths and retirements from injury. After the war, the Fire Service Law was put into effect on 7 March 1948, and city, town, and village fire departments were set up, independently of the police, leading to the current system [32]. With respect to firefighting organisations' records, records of injuries at the time of the atomic bombardment have been kept, but there are no records of the response to the radiation disaster or of relevant education and training from that time to the present day. However, education and training about radiation disasters are currently being provided as an aspect of the training for nuclear-biological-chemical (NBC) disasters.

Of course, not all firefighters in this study are from Nagasaki Prefecture, but they may have some knowledge of atomic bomb and radiation exposure after working in this area. Nevertheless, some participants replied that they had doubts about the value of participating in seminars while firefighters' activities when radiation disasters occur are affected by their level of confidence. On the other hand, Ahayalimudin et al. [15] reported that emergency medical personnel's education level and disaster training attendance was significantly associated with increased knowledge and practice scores and with higher practice scores, respectively. These results suggest it is important to provide a special education programme that focuses on radiation disasters only for firefighters even if they already have some knowledge about radiation exposure.

This study has several limitations. First, among the participants, only one replied that only one indicated having experience in responding to a radiation disaster. It was, therefore, not possible to investigate whether experience with a radiation disaster had any effect on confidence about response to radiation disasters or about handling the relevant materials and/or equipment. Second, the fire departments covered by this study were in Nagasaki Prefecture, and, as there were only three of them, questions about age, gender, and station site were omitted to avoid identifying individuals. If future studies include more firefighters who have experience with radiation disasters and those stationed at fire departments in prefectures that have nuclear power plants, and/or make comparisons between fire departments of approximately equal size in different prefectures, they can be expected to provide additional information that will be useful for making satisfactory preparations for radiation disasters.

The present study found that willingness to accept assignment to a radiation disaster response team was correlated with individual confidence about response to radiation disasters and about handling relevant materials and/or equipment. For this reason, it is considered useful to have continuing educational programmes that combine basic knowledge and practical competence. In the future, the aim is to take measures that reduce anxiety and increase confidence among firefighters regarding response to radiation disasters, including liaison between

fire departments and institutions with specialised knowledge, such as medical centres for people with severe radiation exposure; establishment of seminar programmes specifically for firefighters; and establishment of face-to-face relationships, starting before disasters occur, between firefighting bodies that respond promptly to disasters and other organisations involved with disasters. It is hoped that these measures will facilitate vigorous and proactive activities when disasters strike.

## Supporting information

**S1 File.**
(DOCX)

## Acknowledgments

The authors would like to thank the three fire departments and all personnel who participated in this study. We would also like to thank Dr Maika Nakao for assistance with English proofreading. We would like to thank Editage (www.editage.com) for English language editing.

## Author Contributions

**Conceptualization:** Koji Yamada, Izumi Yamaguchi, Hideko Urata, Naomi Hayashida.

**Data curation:** Koji Yamada.

**Formal analysis:** Koji Yamada.

**Funding acquisition:** Naomi Hayashida.

**Investigation:** Koji Yamada, Naomi Hayashida.

**Methodology:** Koji Yamada, Izumi Yamaguchi, Hideko Urata, Naomi Hayashida.

**Project administration:** Koji Yamada, Naomi Hayashida.

**Supervision:** Izumi Yamaguchi, Hideko Urata, Naomi Hayashida.

**Validation:** Hideko Urata, Naomi Hayashida.

**Writing – original draft:** Koji Yamada, Izumi Yamaguchi.

**Writing – review & editing:** Koji Yamada, Naomi Hayashida.

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
