## [Decision Letter · Decision Letter 0]

23 Apr 2020

PONE-D-19-32209

Survey of awareness about radiation disasters among firefighters in a Japanese prefecture without nuclear power plants

PLOS ONE

Dear Dr. Hayashida,

Thank you for submitting your manuscript to PLOS ONE. After careful consideration, we feel that it has merit but does not fully meet PLOS ONE’s publication criteria as it currently stands. Therefore, we invite you to submit a revised version of the manuscript that carefully and systematically addresses all the points raised by Reviewers 1 and 2 during the review process (see below).

We would appreciate receiving your revised manuscript by Jun 07 2020 11:59PM. To enhance the reproducibility of your results, we recommend that if applicable you deposit your laboratory protocols in protocols.io, where a protocol can be assigned its own identifier (DOI) such that it can be cited independently in the future. For instructions see: http://journals.plos.org/plosone/s/submission-guidelines#loc-laboratory-protocols

We look forward to receiving your revised manuscript.

Kind regards,

Emmanuel Manalo, PhD

Academic Editor

PLOS ONE

Journal Requirements:

1. We note that you have indicated that data from this study are available upon request. PLOS only allows data to be available upon request if there are legal or ethical restrictions on sharing data publicly. For information on unacceptable data access restrictions, please see http://journals.plos.org/plosone/s/data-availability#loc-unacceptable-data-access-restrictions.

Additional Editor Comments (if provided):

Both reviewers note the merits of this paper, but they also point out some very important and necessary modification you need to make before this paper can be considered suitable for publication. So please go through their comments carefully and implement the changes they suggest systematically. I want to emphasize in particular Reviewer 1's comments about the need for a more comprehensive literature review: I myself think that your contextualization of this research within the existing pertinent research literature is very inadequate.

Reviewers' comments:

Reviewer's Responses to Questions

**Comments to the Author**

1. Is the manuscript technically sound, and do the data support the conclusions?

Reviewer #1: Yes

Reviewer #2: Yes

2. Has the statistical analysis been performed appropriately and rigorously? 

Reviewer #1: Yes

Reviewer #2: Yes

3. Have the authors made all data underlying the findings in their manuscript fully available?

Reviewer #1: No

Reviewer #2: Yes

4. Is the manuscript presented in an intelligible fashion and written in standard English?

Reviewer #1: No

Reviewer #2: Yes

5. Review Comments to the Author

Reviewer #1: 1, The authors conducted an important research to facilitate the understanding of the factors affecting individual’s attitudes/decisions towards radiation disaster response. I believe the study has broad impacts, and the results could help develop better disaster preparedness and response. However, the discussion about the intellectual merit of the paper is not enough. How the study would contribute to the existing knowledge of the factors affecting individual’s attitudes/decisions towards disaster preparedness and response is not fully discussed in the paper. I would urge the authors conduct a comprehensive literature review and elaborate the intellectual merits of the paper in both Introduction and Discussion sections.

2. From the lines 145 to 210, the authors explain the survey in detail, and I feel difficult to follow here. The content is helpful for readers to understand the research but also makes the paper tedious. Maybe the authors could find a good way to put it together. For example, the authors could stress some important points in the paper and put more detailed information in the supplemental document.

3. For the chi-square and ordinal logistic regression tests, I would urge the authors either inform the null hypothesis and the alternative hypothesis in the context of study or elaborate what do the results of tests interpret.

4. The discussion related to the results is not sufficient. Also, there are no clear conclusions relevant to the emerging literature. I would urge the authors to revisit some of the literature regarding individual’s attitudes/preferences towards policies of disaster planning and response and discuss how the results could contribute to the existing literature.

5, I suggest that the authors ask for a professional copy editing service. The paper could be greatly improved in terms of the language such as the use of voice, tense, article and the word choice. Also, a common problem is that some sentences are really long and become difficult to follow. The authors could break the long sentences down into several short sentences. I listed some examples that could be improved.

Page 1, line 31-36, line 36-37

Page 2, line 44-49

Page 3-4, line 63-67, line 69 ‘put it out’, line 73-74

Page 5, line 86-90

Page 6, line 97-99

Page 7, line 119-120 and line 121-122 are repetitive.

Page 7, line 121-128 I would suggest that the authors use an active voice instead of a passive voice here.

Reviewer #2: Authors describe a survey of awareness of firefighters for radiation disasters. Results of this survey are important and suggestive for dealing with radiation disasters. However some considerations and improvements in the manuscript will be required.

1. Profile of participants

In this paper, firefighters are classified into 5 categories. Are they engaged in general affairs? I think firefighting department may have the section responsible for special disaster for such as nuclear materials, chemical, biohazards. In this survey, descriptions about firefighters engaged in such works are needed.

2. Significance of survey in Nagasaki “without nuclear plants”

Authors describe Nagasaki as prefecture without nuclear plants. However Nagasaki severely suffered from atomic bomb in 1945. Many people in Nagasaki get intensive education from primary schools. In this point, Nagasaki is different from other prefectures in Japan. Some discussions about this point may be needed.

6. PLOS authors have the option to publish the peer review history of their article (what does this mean?). If published, this will include your full peer review and any attached files.

Reviewer #1: No

Reviewer #2: No

---

## [Author Response · Author response to Decision Letter 0]

6 Jun 2020

We appreciate the time and effort you and each of the reviewers have dedicated to providing insightful feedback on ways to strengthen our paper. We have incorporated changes that reflect the detailed suggestions you have graciously provided. To facilitate your review of our revisions, we replied a point-by-point response to the questions and comments in attached word file named 'response to reviewer'. Could you confirm the attached file? Thank you again.

---

## [Decision Letter · Decision Letter 1]

13 Jul 2020

Survey of awareness of radiation disasters among firefighters in a Japanese prefecture without nuclear power plants

PONE-D-19-32209R1

Dear Dr. Hayashida,

We’re pleased to inform you that your manuscript has been judged scientifically suitable for publication and will be formally accepted for publication once it meets all outstanding technical requirements.

Kind regards,

Emmanuel Manalo, PhD

Academic Editor

PLOS ONE

Additional Editor Comments (optional):

Reviewers' comments:

Reviewer's Responses to Questions

**Comments to the Author**

1. If the authors have adequately addressed your comments raised in a previous round of review and you feel that this manuscript is now acceptable for publication, you may indicate that here to bypass the “Comments to the Author” section, enter your conflict of interest statement in the “Confidential to Editor” section, and submit your "Accept" recommendation.

Reviewer #1: All comments have been addressed

Reviewer #2: All comments have been addressed

2. Is the manuscript technically sound, and do the data support the conclusions?

Reviewer #1: Yes

Reviewer #2: Yes

3. Has the statistical analysis been performed appropriately and rigorously? 

Reviewer #1: Yes

Reviewer #2: Yes

4. Have the authors made all data underlying the findings in their manuscript fully available?

Reviewer #1: Yes

Reviewer #2: Yes

5. Is the manuscript presented in an intelligible fashion and written in standard English?

Reviewer #1: Yes

Reviewer #2: Yes

6. Review Comments to the Author

Reviewer #1: The authors addressed the comments and greatly improved the paper. The research is important because there are few studies regarding firefighters' preparedness for and awareness of regarding radiation disaster response. The paper has great potential and still has large room to be improved in terms of language. One suggestion for author's future writing is to use active voice instead of passive voice in scientific writing. For example, 'A questionnaire survey was carried out with 1,122 firefighters in three firefighting departments in Nagasaki Prefecture, which does not have nuclear power plants. In total, 920 questionnaires were returned, and the 784 that were valid were analysed (page 2, line 25-28)' could be improved as 'We conducted a questionnaire survey with 1,122 firefighters participated in three firefighting departments in Nagasaki Prefecture, where does not have nuclear power plants. In total, we received 920 questionnaires and analysed 784 questionnaires that were valid.' Another suggestion is to avoid very long sentences. For example, 'The 2018 White Paper on Fire Service [6], released after the Fukushima Daiichi Disaster, mentions future measures to be taken based on the disaster, referencing how certain health interventions, for example tests using whole-body counters, were given to firefighters on the emergency firefighting support teams involved in responding to the disaster to help reassure them. (page 7, line 138-142)' I would recommend to accept the manuscript.

Reviewer #2: (No Response)

7. PLOS authors have the option to publish the peer review history of their article (what does this mean?). If published, this will include your full peer review and any attached files.

Reviewer #1: No

Reviewer #2: No

---

## [Editor Report · Acceptance letter]

15 Jul 2020

PONE-D-19-32209R1 

Survey of awareness of radiation disasters among firefighters in a Japanese prefecture without nuclear power plants 

Dear Dr. Hayashida:

I'm pleased to inform you that your manuscript has been deemed suitable for publication in PLOS ONE. Congratulations! Your manuscript is now with our production department. 

Kind regards, 

on behalf of

Professor Emmanuel Manalo 

Academic Editor

PLOS ONE